# Yolk Absorption Rate and Mouth Development in Larvae of *Dormitator latifrons* (Perciformes: Eleotridae)

**Byron Manuel Reyes-Mero** [1,2], **Ana María Santana-Piñeros** [2,*], **Leonela Griselda Muñoz-Chumo** [2], **Yanis Cruz-Quintana** [2] **and Enric Gisbert** [3]

1   Maestría de Investigación en Acuicultura, Instituto de Posgrado, Universidad Técnica de Manabí, Bahía de Caráquez 130104, Ecuador
2   Grupo de Investigación en Sanidad Acuícola, Inocuidad y Salud Ambiental (SAISA), Departamento de Acuicultura, Pesca y Recursos Naturales Renovables, Facultad de Acuicultura y Ciencias del Mar, Universidad Técnica de Manabí, Calle Gonzalo Loor Velasco s/n, Bahía de Caráquez 130104, Ecuador
3   Aquaculture Program, IRTA, Crta. Poble Nou km 5.5, 43540 La Ràpita, Spain
*   Correspondence: ana.santana@utm.edu.ec; Tel.: +593-5-2399300 (ext. 1003)

**Abstract:** Fish larvae suffer high starvation mortality during the transition from yolk absorption to exogenous feeding, and the size of the developing buccal structures limits the food they can consume. Determining the suitable timing and size of live or inert foods could decrease this mortality. We described mouth development and determined the yolk absorption rate and point-of-no-return (PNR) of *Dormitator latifrons* larvae. One male and one female specimen were induced to spawn using salmon GnRHa implants, and 45 of their larvae were sedated and observed under a microscope every 24 h to measure total length, standard length, yolk sac length, yolk sac width, oil globule length, oil globule width, width of the oesophagus, and length of the upper and lower jaw longitudinal. The growth model, maximum mouth opening, daily survival, and starvation period were determined. The larval growth was fastest during the first 24 h post−hatching (HPH) at 28 ± 1 °C with an average increase of 625.42 μm in total length and 573.51 μm in standard length. The highest percentage of yolk absorption (52%) occurred within 24 HPH and at 96 HPH the yolk sac was completely reabsorbed. The PNR was reached at 156.41 HPH ($p < 0.05$). At 96 HPH, the upper and lower jaw were distinguishable by mouth movements. Our results suggest that the larvae of *D. latifrons* should be fed at 96 HPH with prey measuring 50–65 μm.

**Keywords:** Pacific fat sleeper; morphology; larvae culture; eastern central Pacific; exogenous feeding

## 1. Introduction

A critical point in the development of fish larvae is finding adequate food items during the transition from the lecithotrophic to the exogenous feeding stage. The stage at which the yolk is exhausted has massive mortality if proper food items are not available [1,2]. Short periods of food deprivation after yolk resorption can result in abnormal behaviour and morphological development, degeneration of the alimentary tract and trunk musculature, and reductions in food utilisation efficiency and feeding activity, as well as impacting larval mortality. This mortality, due to the progressive loss of larval energy reserves, is reduced when exogenous feeding is appropriate and of sufficient quality [3].

Knowing the age or size when first feeding normally occurs is particularly important, as is knowing how long larvae can withstand food deprivation before reaching the point-of-no-return (PNR) when the cumulative effects of starvation become irreversible, and 50% of starved larvae are still alive but are unable to feed even when food resources become available [4]. In culture conditions, the PNR is important, as it allows the synchronisation of the rearing process to the physiological state of the larvae, thus, maximising larval survival and growth, as well as minimising size variation, whereas in natural environments, this information is key for understanding larval recruitment [5].

Species of the genus *Dormitator* (Gill, 1861), also known as sleepers, are amphidromous freshwater fish that inhabit the tropical and subtropical regions of the Americas and western Africa [6]. *Dormitator latifrons* (Richardson, 1844), or the Pacific fat sleeper, is native to the eastern central Pacific and inhabits coastal lagoons, mangroves, and estuaries [7]. *D. latifrons* is a very versatile species that can be kept in freshwater (0 PSU) and marine water (<40 PSU) [8]. In natural conditions, *D. latifrons* spawn in freshwaters bodies (0–5 PSU), and the larvae migrate to the sea and then return to freshwater to spend the rest of their development [9,10]. This species is produced in Ecuador in small−scale systems where productions range from 800 to 1000 t [11]. *D latifrons* farming currently consists of fattening juveniles taken from natural environments, which has limited the development of its culture, as well as calling into question the sustainability of this practice [11]. The producers and anglers of *D. latifrons* mention the need to develop larviculture protocols for this species, since production volumes cannot be maintained using wild−caught fingerlings due to natural fluctuations in their abundance [8]. In addition, Gonzalez-Martinez et al. [7] mention that these practices have resulted in a reduction in the size of the fish and a loss of genetics due to the overexploitation of *D. latifrons*. Thus, it is expected to develop research in larviculture and manage programs for the productive development and conservation of the species, given the impact of the current farming model for the species [7].

Previous research on *D. latifrons* achieved the release of viable gametes using gonadotropin-releasing hormones (GnRHa and LHRHa) in experimental conditions at low salinity (0–5 PSU) [12], although these authors mention that the fertilisation process can also occur with a salinity of up to 15. The digestive and visual tracts of larvae have been described up to day 6 post−hatching [13]. Additionally, Reyes et al. [14] tested the exogenous feeding of *D. latifrons* larvae on live *Proales similis* de Beauchamp, 1907, a rotifer species measuring approximately 100 μm; however, the larvae did not survive after 7 days post−hatching. The size of the opening of the mouth is the main limitation for the feeding of larval fish [15], making food particle size one of the crucial factors for the survival of larval stages. To determine the ideal size of the particles supplied as exogenous food during the first days of life of *D. latifrons*, the objectives of this study were to describe the development of the mouth and determine the yolk absorption time and point-of-no-return (PNR).

## 2. Materials and Methods

### 2.1. Sample Collection

Ten adults of *D. latifrons* (5 females and 5 males) were collected in the La Segua wetland (0°42′33″ S—80°11′54.43″ W), Manabí Province, Ecuador. La Segua wetland harbours the growth area of *D. latifrons* and it is located at the confluence of the Carrizal and Chone rivers. It includes a central permanent freshwater lagoon (up to 2 PSU) and a wide plain seasonal flooded area [16]. The five females had an average weight of 762 g (742–777 g) and an average total length of 33 cm (32.5–33.5 cm), and the five males had an average weight of 836 g (816–898 g) and average total length of 35.2 cm (35–35.5 cm). Fish were transferred to the facilities of the Sucre Extension, Universidad Técnica de Manabí, where they were acclimatised for 3 weeks in tanks with 200 L of freshwater (2 PSU) at water temperature (27 to 29 °C), photoperiod of 12 h light:12 h dark, pH of 8.0 to 8.5, ammonia concentration 0.79 mg/L, dissolved oxygen 4 mg/L, oxygen saturation 90%, aeration supplied with diffuser stones, and partial exchange of water was performed daily, during daytime. The specimens were fed at 2% of biomass per day with balanced shrimp feed (22% protein, Feedpac® growth, Agripac, Guayaquil, Ecuador).

### 2.2. Spawning and Larval Rearing

One male and one female specimen with the most evident reproductive characteristics (colour intensity, bulky belly, and larger size) were sedated with 2-phenoxy−ethanol diluted in fresh water (0.7 mL/L) for 2 to 3 min. Subsequently, these specimens were selected and isolated in a 200 L tank and were induced to spawn using Atlantic salmon GnRHa implants [12]. The eggs were not counted because some of them adhered to

the surface of the tank, and the others kept floating. The larvae were counted with the traditional volumetric method. A total of 953 550 larvae hatched 7 to 8 h after fertilisation. Subsequently, brood stock was returned to the tanks of origin. Forty-five larvae were sedated once every 24 h with 2−phenoxy−ethanol diluted in fresh water and observed under a binocular microscope. Images were taken with an 18 MP camera attached to a microscope. Image analysis and measurements were performed using the program ImageJ [16]. The measurements taken from each larva were the following: total length (TL), standard length (SL), yolk sac length (YSL), yolk sac width (YSW), oil globule length (OGL), oil globule width (OGW), the width of the oesophagus (WE), and length of the upper and lower jaw (UJL and LJL) (Figure 1).

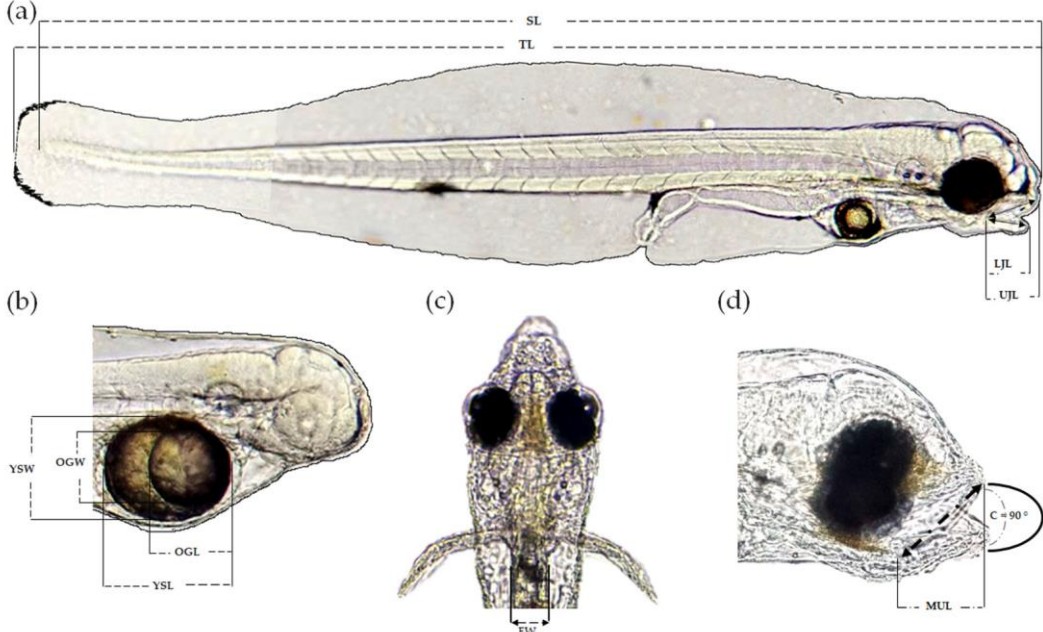

**Figure 1.** Morphometry of Pacific fat sleeper larvae *Dormitator latifrons*; (**a**) total length (TL), standard length (SL), upper jaw length (UJL), and lower jaw length (LJL); (**b**) yolk sac length (YSL), yolk sac width (YSW), oil globule length (OGL), and oil globule width (OGW); (**c**) width of the oesophagus (WE); (**d**) maximum length of the upper jaw considering an angle of 90° (MUL).

*2.3. Data Analysis*

Linear regression analysis was performed to determine the growth pattern of the larvae. To calculate the ellipsoidal volume of the yolk sac and the oil globule, the methodology of Blaxter and Hempel [4] was used with the following Equation (1):

$$V = \frac{\pi}{6} L \times W^2 \tag{1}$$

where $V$: volume expressed in mm$^3$;

$\quad$ $L$: total length expressed in mm;

$\quad$ $W$: width expressed in mm.

To estimate the maximum mouth opening (*MMO*), the methodology of Shirota [17] was used, which considers an angle of 90° through the following Equation (2):

$$MMO = MUL \times \sqrt{2} \tag{2}$$

where *MUL* is the maximum upper length of the maxilla in μm (Figure 1).

The determination of the point-of-no-return (PNR) was calculated from the starvation period through the daily survival of the larvae [4]. The starvation period was estimated using the response time analysis proposed by Kappenberg and Rahnenführer [18], using R

Studio software, version 4.2.0 (R Core Team, 2022). Data are shown as mean $\pm$ standard deviation.

## 3. Results

### 3.1. Larval Growth

The greatest increase in TL and SL is observed during the first 24 HPH, with an average increase of 625.42 μm in TL and 573.51 μm in SL (Figure 2A,B,I,J). From 48 to 168 HPH, the growth rate in TL and SL is approximately 2.94%/day and 2.90%/day, respectively (Figure 2C–H). Larval growth was fitted to a linear model ($R2 = 0.50$; $p < 0.05$) with the equation $TL = 1361.44 + 3.3t$, where TL is the total length and $t$ is the time in hours post−hatching.

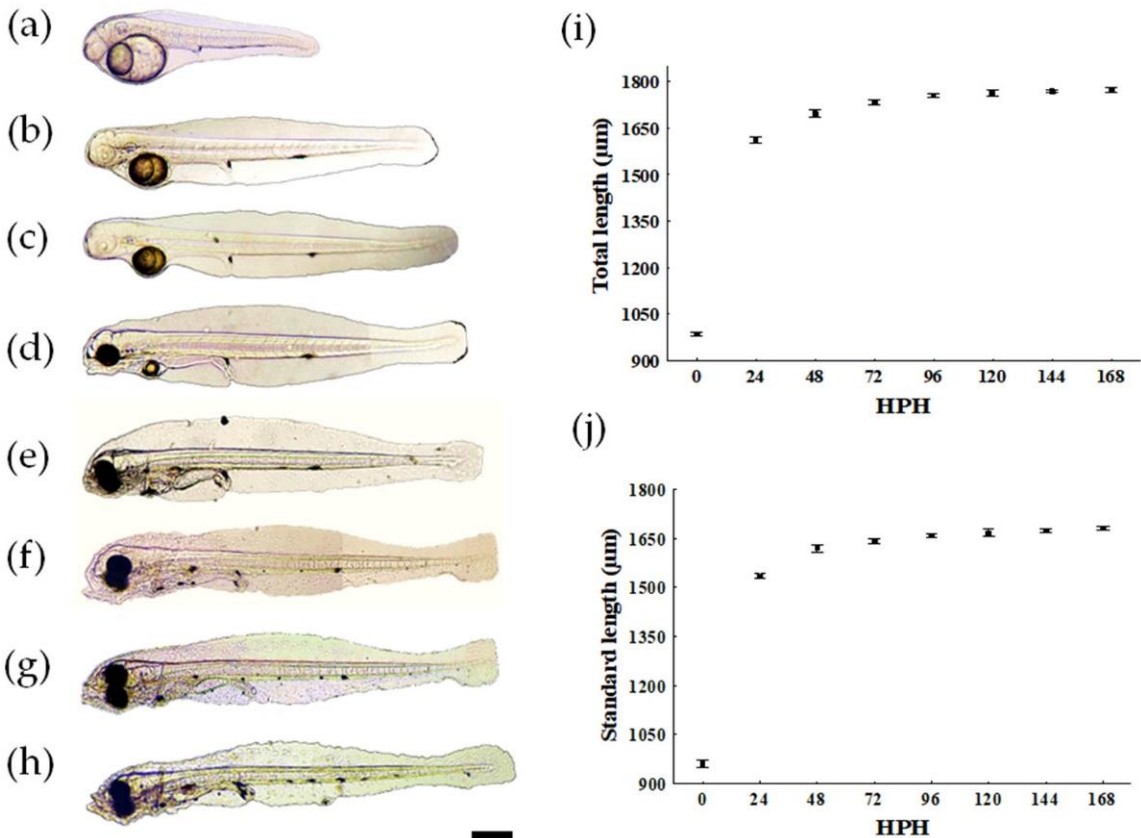

**Figure 2.** Larval growth of Pacific fat sleepers *Dormitator latifrons*; (**a**) 2 h post−hatch (HPH), (**b**) 24 HPH, (**c**) 48 HPH, (**d**) 72 HPH, (**e**) 96 HPH, (**f**) 120 HPH, (**g**) 144 HPH, and (**h**) 168 HPH; (**i**) total length, (**j**) standard length; HPH: hours post−hatch.

### 3.2. Yolk Absorption

Newly hatched larvae (0 HPH) have a yolk longitudinal axis diameter of 287.58 $\pm$ 19.49 μm. An oil globule (g) is observed, which occupies 42% of the yolk (y) (Figure 3A). The highest percentage of yolk absorption (52%) occurs within 24 HPH and the size of the oil globule decreases by 13% compared to its size at 0 HPH (Figure 3B). At 48 and 72 HPH, there is a constant and gradual decrease in the yolk (approx. 15%) (Figure 3C–E), while the oil globule decreases considerably (45%) by 72 HPH (Figure 3F). At 96 HPH, the yolk is completely absorbed. Response time analysis estimated the PNR in *D. latifrons* larvae at 156.41 HPH (ca. 6.5 days) at 28.0 $\pm$ 1.0 °C ($p < 0.05$).

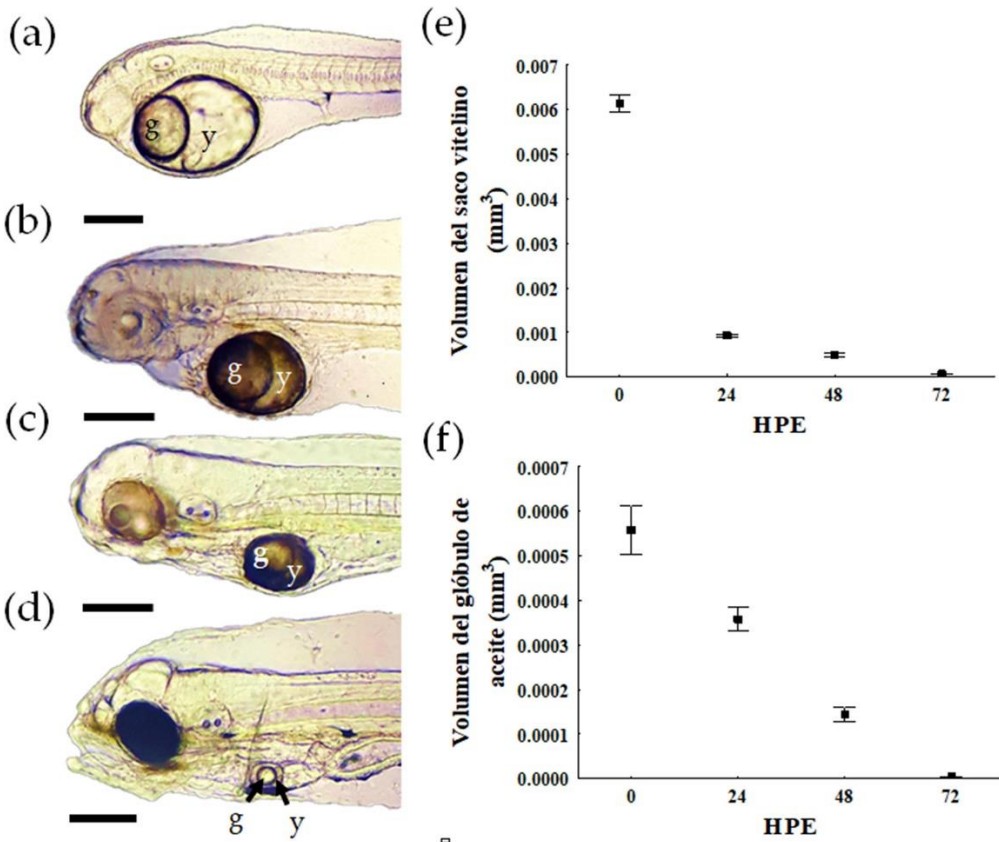

**Figure 3.** Yolk absorption of larval Pacific fat sleepers *Dormitator latifrons*; (**a**) 2 h post−hatch (HPH), (**b**) 24 HPH, (**c**) 48 HPH, and (**d**) 72 HPH; (**e**) longitudinal axis of the yolk sac as a function of HPH, (**f**) longitudinal axis of the oil globule versus day post−hatching; y: yolk; g: oil globule; scale = 100 μm.

### 3.3. Mouth Morphometry

At 0 and 24 HPH, the mouth is closed (Figure 4A,B). At 48 HPH, the mouth is still closed, but we observe the beginning of the formation of the oral cleft (Figure 4C). At 72 HPH, the mouth is open, and the lips begin to form (Figure 4D). At 96 HPH, the upper jaw and the lower jaw can be distinguished, with the upper jaw larger than the lower jaw (Figure 4E). Moreover, the mouth begins to show movement and opens to a maximum height of 66.13 ± 7.28 μm (Table 1), and peristaltic movements and the melanin plug are observed in the digestive tract. At 96 and 120 HPH, the upper jaw and lower jaw have their largest average size increase of 52 μm/day (Figure 4E,F). At 120 HPH, the upper jaw has a shorter length than the lower jaw, maintaining this trend until 168 HPH. After 144 HPH, the growth of the upper jaw and lower jaw slows compared to the previous days (Figure 4G; Table 1). The average maximum mouth opening at 168 HPH is 217.17 ± 33.96 μm (Figure 4H; Table 1). The average width of the oesophagus from 72 HPH to 168 HPH is 69.35 ± 11.68 μm (Figure 4I; Table 1).

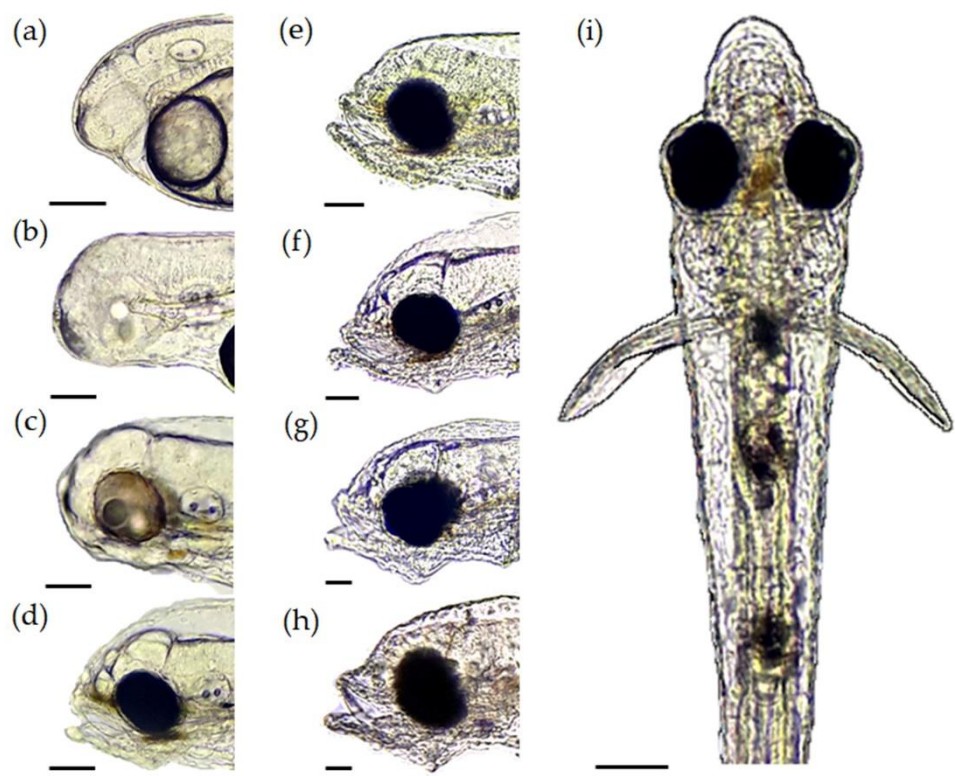

**Figure 4.** Larval mouth development of Pacific fat sleepers *Dormitator latifrons*; (**a**) 2 h post−hatch (HPH), (**b**) 24 HPH, and (**c**) 48 HPH; opening of the buccal cavity in larvae at 72 HPH, (**d**) 96 HPH, (**e**) 120 HPH, (**f**) 144 HPH, (**g**) and 168 HPH; (**h**) scale a–h = 50 µm. (**i**) Dorsal view of larvae at 96 HPH; scale = 100 µm.

**Table 1.** Morphometry of the mouth development of *Dormitator latifrons* larvae from 0 h post-hatching (HPH) to 168 HPH values are presented as mean ± standard deviation (*n* = 45).

| Age | Jaw Length (µm) | | Maximum Mouth Opening (µm) | Width of the Oesophagus (µm) |
|---|---|---|---|---|
| | Upper | Lower | | |
| 0 | Closed mouth | Closed mouth | Closed mouth | Closed mouth |
| 24 | Closed mouth | Closed mouth | Closed mouth | Closed mouth |
| 48 | Closed mouth | Closed mouth | Closed mouth | Closed mouth |
| 72 | 43.22 ± 5.12 | 40.72 ± 4.96 | 61.13 ± 7.28 | 53.87 ± 6.65 |
| 96 | 95.83 ± 12.15 | 88.48 ± 13.34 | 135.53 ± 17.69 | 65.67 ± 4.04 |
| 120 | 147.45 ± 24.54 | 160.06 ± 23.79 | 208.54 ± 34.70 | 80.88 ± 8.30 |
| 144 | 148.70 ± 24.54 | 164.88 ± 26.26 | 210.29 ± 34.70 | 72.12 ± 8.42 |
| 168 | 153.56 ± 24.02 | 174.50 ± 25.14 | 217.17 ± 33.96 | 74.19 ± 8.44 |

## 4. Discussion

*Dormitator latifrons* larvae grow quickly during the first 24 HPH, then grow more slowly until 72 HPH, after which the growth rate further decreases. The growth of the larvae during the first 72 HPH coincides with the period of the highest consumption of yolk, approximately 80%, during this period. The lack of growth after 72 HPH could be due to starvation during this transitory period of lecithotrophy to exogenous feeding. It is important to mention that no larviculture protocol has been developed so far, and researchers working on this species have not been able to find the optimal feeding protocol in terms of the type or live prey, size, and nutritional value. Growth patterns of *D. latifrons* larvae coincide with the report of Farris [19], which divided larval growth into three phases: the rapid growth stage immediately after hatching, the slow growth stage that occurs during the period of yolk depletion, and the negative growth stage that occurs after failure to establish. Furthermore, Auty [20] and Llewellyn [21] report comparable results in the

eleotrids *Hypseleotris compressa* (Krefft, 1864) and *Awaous melanocephalus* (Bleeker, 1849), where the larvae survive for 96 h after consumption of the yolk at 28.0 ± 1.0 °C. Although López-López et al. [13] and Rodíguez-Montes de Oca et al. [12] managed to produce *D. latifrons* larvae up to 4 and 6 DPH, respectively, they do not mention the growth rate of the fingerlings, or the rate of yolk decrease during this period. Therefore, our results constitute the first evidence of the growth of *D. latifrons* larvae during the lecithotrophic stage under controlled experimental conditions.

The absorption times of the yolk and oil globule in this study are observed at 96 HPH at 27 °C, which differs from the results of López-López et al. [13] and Rodríguez-Montes de Oca et al. [12], in which the yolk is absorbed between 72 and 96 HPH and 72 HPH at 26 °C, respectively. Although higher temperatures increase basal metabolism, accelerating the absorption of the yolk sac and the development of the larva, the absorption time of YS in our study with a slightly higher temperature (1 °C) is higher than reported by López-López et al. [13] and Rodríguez-Montes de Oca et al. [12], probably due to the larval size at hatching. Our yolk measurements are 2.12 times the value reported by Rodríguez-Montes de Oca et al. [12], which probably lengthened the yolk absorption time in this study. The spawners we induced are larger (742 g in females and 816 g in males) than those used by Rodríguez-Montes de Oca et al. [12] (486 g in females and 623 g in males), while López-López et al. [13] do not mention sizes or weights of organisms used. Patimar et al. [22] found that *Cobitis satunini* Gladkov, 1935 fecundity and egg size increases linearly with increasing spawner size, which could explain the difference between the studies. Another possible explanation is the phenotypic traits of the spawner populations used. Butts and Litvak [23] indicated that the size and growth rate of the larva can vary depending on the origin of the spawners. In this study, the broodfish came from wild populations in Ecuador, while the spawners used by Rodríguez-Montes de Oca et al. [12] and López-López et al. [13] were from Mexico.

The transition period between the absorption of yolk and exogenous feeding in *D. latifrons* larvae lasts for 60 h; therefore, that the larvae must seek, capture, and assimilate food before entering the period of irreversible starvation, the so−called PNR, which is determined for this study at 156 HPH. Several authors mentioned that during this period, larvae become weak and die despite having food available, since they are unable to feed in their weakened state [24,25]. In this study, mass mortality of *D. latifrons* larvae is observed at 168 HPH (7 DPH), which coincides with the period of mass mortality reported in other eleotrids such as *Oxyeleotris marmorata* (Bleeker, 1852) (8 DPE, 27 °C) [26] and *Hypseleotris compressa* (10 DPE, 30 °C) [20]. Dou et al. [27] established that to reduce mortality due to starvation, appropriate exogenous food should be added before the yolk is absorbed. According to our results, we suggest *D. latifrons* larvae be fed from 72 HPH and not later than 96 HPH at 28.0 ± 1.0 °C to avoid irreversible damage to their development.

The mouth of the larvae of *D. latifrons* begin to show movements beginning at 96 HPH, which coincides with the observation of peristaltic movements and evacuation of the melanin plug in the digestive tract. This result agrees with our initial hypothesis of providing live prey from 72 HPH, since the larvae must learn to capture prey even though their mouths are still poorly developed. Similarly, López-López et al. [13] observe the ingestion of particles of diet offered from 96 HPH in *D. latifrons* larvae, and histological analyses show that the visual system shows advanced development at 96 HPH, achieving full development at 120 HPH. The evacuation of the melanin plug has been suggested as the initial point of exogenous feeding [28].

At 72 HPH, we observe that the buccal opening still has poorly developed mouth structures, but by 96 HPH, the buccal opening has doubled in height, the upper jaw and lower jaw are distinguishable, and the width of the oesophagus has increased (66 μm). According to these results, we suggest evaluating the introduction of live prey at 72 HPH, with live prey sizes ranging from 50 to 65 μm, and progressively increasing the prey size to 80 μm by 96 HPH. Yúfera and Darias [3] mentioned that the choice of prey is limited by the size of the mouth and the diameter of the oesophagus. Smaller prey sizes are preferable,

since the larvae swallow the prey whole, and the diameter of the oesophagus defines the maximum size that can be ingested. The deterioration of the larva due to starvation is observed between 120 HPH and 144–168 HPH, during which is was a decrease in the average width of the oesophagus. Several authors mention that during the food absence period, changes are observed in organs and tissues, which leads to growth, stagnation, and thinning of the intestinal walls [5,29,30].

In conclusion, the present study indicates that the PNR in *D. latifrons* is at 156 HPH when reared at 28.0 ± 1.0 °C. In addition, live prey administration is recommended before the complete resorption of the yolk sac at 96 HPH, with live prey sizes comprised between 50 and 65 μm, whereas their size should increase progressively up to 80 μm by 96 HPH. These results are the first step in the development of a larvae culture protocol for this species, where the transition from the lecithotrophic stage to exogenous feeding still represents the main challenge for the larviculture of this eleotrid species.

**Author Contributions:** Conceptualization: A.M.S.-P. and Y.C.-Q.; data curation: B.M.R.-M. and L.G.M.-C.; formal analysis: B.M.R.-M., L.G.M.-C. and A.M.S.-P.; funding acquisition: A.M.S.-P., Y.C.-Q. and E.G.; investigation: A.M.S.-P., B.M.R.-M. and Y.C.-Q.; methodology: A.M.S.-P., B.M.R.-M., L.G.M.-C. and Y.C.-Q.; resources: A.M.S.-P. and Y.C.-Q.; supervision: A.M.S.-P. and Y.C.-Q.; validation: A.M.S.-P., Y.C.-Q. and E.G.; writing—original draft: B.M.R.-M. and A.M.S.-P.; writing—review and editing: A.M.S.-P., Y.C.-Q. and E.G. All authors have read and agreed to the published version of the manuscript.

**Funding:** This research received no external funding.

**Institutional Review Board Statement:** The specimens were collected under a collection permit (No. 004-2019-DP-DPAM-MAE) issued by the Ministry of the Environment (Ministerio de Ambiente, Ecuador) of Ecuador.

**Informed Consent Statement:** Not applicable.

**Data Availability Statement:** Not applicable.

**Acknowledgments:** We are grateful to the Universidad Técnica de Manabí for financial support through the project "Aspectos biológicos del chame *Dormitator latifrons* en ambientes naturales y de producción"; to Centro de Sanidad Acuícola of the Departamento de Acuicultura, Pesca y Recursos Naturales Renovables; to the artisanal fishermen of the La Segua wetland. Collaboration between Ibero-American researchers was performed under the framework of the network LARVAplus "Strategies for the development and improvement of fish larvae production in Ibero-America" (117RT0521) funded by the Ibero-American Program of Science and Technology for Development (CYTED, Spain).

**Conflicts of Interest:** The authors declare no conflict of interest.

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
