# Peer review of "Yolk Absorption Rate and Mouth Development in Larvae of Dormitator latifrons (Perciformes: Eleotridae)"

_fishes, doi:10.3390/fishes7060375_

Round 1
Reviewer 1 Report
Overall the study „Yolk absorption rate and mouth development in larvae of Dormitator latifrons (Perciformes: Eleotridae)“ can be interesting to a broader range of readers. It has the potential to improve the rearing of the species and by this to ease pressure on wild populations. The study design is simple but adequate and the results can allow practical applications to be adapted to improve procedures on the species aquaculture husbandry. However, some points need to be improved before the publication can be accepted.
The major point of critique is the low number of compared studies, which should be improved to give a broader view of the research status of this species and its development. Also, the environment chosen for rearing needs to be described in more detail. Following some points for improvements are mentioned and major points are marked with (!).
Good luck with the manuscript
Introduction:
Line 34: Maybe use a dot instead of a semicolon.
Line 50: Please also write the source of the first describer for the genus Dormitator in brackets as well.
Line 55: Please use the correct abbreviations for tons (t).
Line 55: Please shorten the genus name as it has been used in this chapter before already.
Line 58: Please remove the word larvae as it is by sense already included in larviculture (or alternatively larval culture?).
Materials and Methods:
2.1: The averages could be alternatively given with the standard deviation in the form of „average±SD unit“. Otherwise, maybe add the units after the min-max named in the brackets.
(!) 2.1: Please be more specific about the raising environment parameters, for example, add information on the light cycle duration and how the aeration was achieved (influence on water flow velocity).
(!) Line 84-85: Is it appropriate to use freshwater for this brackish water species? Do 2 UPS mean 2 PSU (Practical Salinity Unit) maybe?
Fig.1 Legend: Standard length is here abbreviated as SL. Please make it consistent throughout the manuscript. Generally, I think it is more common to use the abbreviation SL instead of ST in the literature. Maybe change this abbreviation overall?
(!) How where the number of larvae counted? (Line 94-95) Please clarify.
Results:
Line 126: SL is used for the standard length here. (see before comment)
Fig. 2: Would it be possible to adapt the figure to better display the variability between the specimens of the same age stage (like a scale adaption)? The present form does not allow for distinguishing the in-stage differences in size sufficiently (compare to Fig. 3 with a better variance depiction).
Discussion:
(!) Line 188-205: Were the temperatures for the raise given in the mentioned studies? Temperature is known to affect growth and metabolism strongly, thus also yolk absorption rate, and should be considered here as well. The mentioning of environmental parameters in lines 204-205 is not sufficient for this. Comparisons could also be made based on degree-day units.
Sources:
(!) Overall, the manuscript lacks comparisons with other previous studies. Already a short check on Google Scholar gave out further studies, which should be included. Exemplarily:
· Chang, B. D., & Navas, W. (1984). Seasonal variations in growth, condition and gonads of Dormitator latifrons (Richardson) in the Chone River Basin, Ecuador. Journal of Fish Biology, 24(6), 637-648.
· Basto-Rosales, M. E. R., de Oca, G. A. R. M., Carrillo-Farnés, O., Álvarez-González, C. A., Badillo-Zapata, D., & Villasante, F. V. (2019). Growth of Dormitator latifrons under different densities in concrete tanks. Tropical and Subtropical Agroecosystems, 22(2).
· Chang, B. D. (1984). Tolerances to salinity and air exposure of Dormitator latifrons (Pisces: Eleotridae). Revista de Biología Tropical, 32(1), 155-157.
· Morelos‐Castro, R. M., Román‐Reyes, J. C., Luis‐Villaseñor, I. E., Ramírez‐Pérez, J. S., & Rodríguez‐Montes de Oca, G. A. (2020). Expression analyses of digestive enzymes during early development and in adults of the chame fish Dormitator latifrons. Aquaculture Research, 51(1), 265-275.
Author Response
We welcome reviewer comments. Attached comments to reviewer 1.
Reviewer 2 Report
In this manuscript entitled “Yolk absorption rate and mouth development in larvae of Dormitator latifrons (Perciformes: Eleotridae)”, authors described ontogenetic process of mouth development and yolk absorption and determined PNR of D. latifrons. Authors revealed the timing of mouth opening and yolk absorption and indicated that PNR of this fish was at 156 HPH. From these results, authors suggest that this fish larvae should be fed at 96 HPH. This study provides useful information for developing artificial seedlings of D. latifrons larvae. There are no serious technical problems in this study, and thus it is considered worthy of publication in Fishes. However, the following points need to be improved.
Major comments
Although there is no information on feeding for larvae in this article, authors mentioned that they observed food particles in the digestive tract in discussion. If you are feeding, this should be noted in the manuscript. If you are not feeding, you explain what the food particles are.
Minor comments
Unify the abbreviation of width of the esophagus. L101 EW, L107 WE
Author Response
We welcome reviewer comments. Attached comments to reviewer 2.